# Diagnostic Performance of Ex Vivo Fluorescence Confocal Microscopy in the Assessment of Diagnostic Biopsies of the Prostate

**DOI:** 10.3390/cancers13225685

**Published:** 2021-11-13

**Authors:** Ulf Titze, Torsten Hansen, Christoph Brochhausen, Barbara Titze, Birte Schulz, Alfons Gunnemann, Bernardo Rocco, Karl-Dietrich Sievert

**Affiliations:** 1Institute of Pathology, University Hospital OWL of the University of Bielefeld, Campus Lippe, 32756 Detmold, Germany; Torsten.Hansen@klinikum-lippe.de (T.H.); barbara.titze@klinikum-lippe.de (B.T.); birte.schulz@klinikum-lippe.de (B.S.); 2Institute of Pathology, University of Regensburg, 93053 Regensburg, Germany; Christoph.Brochhausen@klinik.uni-regensburg.de; 3Central Biobank Regensburg, University Hospital Regensburg, 93053 Regensburg, Germany; 4Department of Urology, University Hospital OWL of the University of Bielefeld, Campus Lippe, 32756 Detmold, Germany; alfons.gunnemann@klinikum-lippe.de (A.G.); kd_sievert@hotmail.com (K.-D.S.); 5Department of Urology, University of Modena and Reggio Emilia, 41124 Modena, Italy; bernardo.rocco@gmail.com

**Keywords:** confocal microscopy, prostate cancer, digital pathology

## Abstract

**Simple Summary:**

Fluorescence confocal microscopy (FCM) is a novel micro-imaging technique providing optical sections of examined tissue. In this study, we compare intraoperative diagnoses from the real-time application of FCM in pre-therapeutic prostate biopsies with the final diagnoses from conventional histology. We found FCM to be an effective tool for the timely assessment of prostate biopsies enabling reliable real-time diagnosis of prostate cancer in patients requiring therapy.

**Abstract:**

Background: Fluorescence confocal microscopy (FCM) is a novel micro-imaging technique providing optical sections of examined tissue. The method has been well established for the diagnosis of tumors in dermatological specimens. Methods: We compare intraoperative diagnoses of the real-time application of FCM in pre-therapeutic prostate biopsies (35 patients, total number of biopsy specimens: n = 438) with the findings of conventional histology. Results: Prostate carcinoma was reliably diagnosed in all patients. Depending on scan quality and experience of the examiner, smaller lesions of well differentiated carcinoma (ISUP1) could not be consistently differentiated from reactive changes. Furthermore, in some cases there was difficulty to distinguish ISUP grade 2 from ISUP grade 1 tumors. ISUP grades 3–5 were reliably detected in FCM. Conclusions: Despite some limitations, FCM seems to be an effective tool for the timely assessment of prostate biopsies enabling reliable diagnosis of prostate cancer in patients requiring therapy.

## 1. Introduction

Prostate carcinoma (PCa) is the most common non-skin cancer in men in the western world [1]. The highest incidence is found in the United States (124.8/100,000), especially among African American men (185.4/100,000). In Germany, PCa accounts for 25.4% of all diagnosed cancers, which corresponds to 60,000 new cases per year [2].

PCa has long been known to be a heterogeneous disease from a clinical and morphological perspective [3]. The clinical presentation can range from localized indolent to a rapidly progressing lethal metastatic disease. Although the majority of men are diagnosed with organ-confined tumor, long-term outcomes can vary greatly [4]. The gold standard for this diagnosis is the histological evidence of malignancy in sonographic guided or MRI-fused biopsies. Risk stratification and therapeutic decisions are currently based on the clinical examination, the serum PSA and the histological grading according to GLEASON/ISUP from prostate biopsies [5]. Over the past decade, extensive profiling studies delivered new insights of the molecular and phenotypic complexity of primary and metastatic PCa [6]. It can be estimated that clinical challenges posed by the complex heterogeneity will continue to require multi-disciplinary approaches, including novel computational techniques and deep learning algorithms to analyze the resulting multi-dimensional data [7].

Ex vivo fluorescence confocal microscopy (FCM) is a digital micro-imaging technique that provides optical sections of unfixed tissue [8]. The images closely resemble frozen sections allowing the timely histological diagnoses from surgical specimens and biopsies without any loss of tissue. The method is established in dermatology for the routine diagnostics of neoplastic and inflammatory skin diseases [9]. Preliminary investigations of other various organs tumors have already been published and show promising results [10,11]. First results from a European prospective multi-center study recently compared diagnoses of PCa based on FCM-images with conventional histological processing as the recent gold standard diagnostics. Acquisition of FCM images provided reliable diagnoses of PCa in real-time offering opportunities for immediate sharing and reporting from remote pathologists [12,13,14].

In this prospective study, we present our practical experience in the application of FCM for the assessment of prostate biopsies. The primary endpoint of our investigations was the agreement of cancer diagnoses based on FCM and conventional histology. Secondary endpoints were the levels of inter-observer variability and the ability to distinguish between ISUP grade 1 and ISUP grade > 1 cancers based on the FCM scans.

## 2. Materials and Methods

### 2.1. Study Participants

MRI-fused prostate punch biopsies from 35 patients between 49 and 79 years (mean 65.7 ± 7.8) were examined. Six patients were under active surveillance with previously known prostate cancer, the other patients were clinically suspected of having prostate cancer. Before the biopsy was taken, the study participants were informed about the examinations and written consent was obtained from all participants. The study was approved by the ethics committee of the Westphalia-Lippe Medical Association (file number 2020-029-f-S) and carried out in accordance with the ethical principles of the Declaration of Helsinki.

### 2.2. Study Design

MRI-fused biopsy cylinders (targeted and random) were taken from each patient. Depending on imaging and clinical presentation, variable numbers of biopsy cylinders were examined in real-time with the confocal microscope in the operating room. The FCM scans were blindly evaluated by two experienced pathologists (UT, BT). Conventional histology was performed by two other experienced pathologists: A third pathologist at our facility performed the routine diagnostics (TH) and a fourth pathologist (ChB) from an external facility blindly re-examined the HE-sections. In case of discrepant histological diagnoses, consensus was established for further analysis. Finally, we statistically compared the findings from FCM to the final results from conventional histology.

### 2.3. Acquisition of FCM-Images and Sample Processing

According to the manufacturer’s instructions, the selected native biopsy cylinders were pre-treated with pure alcohol for 10 s (protein precipitation for contrast enhancement). The tissue was then incubated for 30 s with an Acridine Orange solution (AO, 0.6 mM; Sigma-Aldrich^®^, St. Louis, MO, USA). AO is an intercalating fluorochrome for staining nucleic acids, whose DNA -dye complexes have an absorption maximum of 502 nm and an emission maximum of 526 nm [15].

The tissue samples were placed on foam pads and then positioned between two microscope slides specially modified with magnets. This temporary specimen was inserted into the specimen holder of the confocal microscope and the scan was performed. The confocal scans were obtained in the operating room by the attending pathologist (UT) with experience in uropathology and diagnoses were established in real-time. Blinded re-evaluation of the FCM-scans (BT) for this study was performed several weeks after the clinical cases were closed.

After the scan was made, the tissue was immediately placed into labelled embedding capsules and fixed in 4% PBS-buffered formaldehyde for 24 h. The further histological investigation was carried out immediately for all biopsies following the standard procedure for formalin-fixed and paraffin embedded (FFPE) tissue. The biopsies were not removed from the foam pads during further processing in order not to change their orientation. In this way, the FCM scan could be directly compared with the histological slides of the same tissue. Minor differences in the structures shown in the figures at higher magnifications result from the different penetration depths of the compared methods (FCM: 30–100 µm vs. up to 16 serial sections in conventional histology resulting in penetrations depths of 1000 µm and more). 

### 2.4. Histological Evaluation

Histological diagnoses were established based on eight serial HE-stained sections from each of the paraffin-fixed biopsies. In case of doubt, immunohistological stains for p40 and AMACR (both polyclonal antibodies, Zytomed Systems, Berlin, Germany) were performed using the Ventana Benchmark^TM^ platform (Ventana Medical Systems, Tucson, AZ, USA). Diagnoses for routine histology (pathologist TH) were established on the next day according to the current guidelines in order to avoid any delays in patient management. Blinded re-evaluation for this study (pathologist ChB) took place several weeks after conventional histology was completed.

### 2.5. Ex Vivo Fluorescence Confocal Microscopy

The FCM-examinations were carried out with a laser scanning confocal microscope of the type VivaScope^®^ 2500 M-G4 from Mavig, Munich, Germany. Illumination of the specimen was performed by two lasers with wavelengths of 488 nm (ultraviolet) and 785 nm (near-infrared). The short-wave laser represented the cell nuclei marked with fluorescence-dye before the examination. The cytoplasmic and extracellular structures were recorded by the reflected light of the long-wave laser. A built-in algorithm transformed the recorded gray values into an image similar to hematoxylin-eosin (HE) staining, in which the nuclei of the cells were shown in violet, whereas connective tissue fibers and cytoplasm of the cells were shown in pink [16]. By modulating the intensity of the two lasers, staining intensity of nuclear and cellular structures could be modified independently. The microscope was equipped with a water immersion objective with 38× magnification and a numerical aperture of 0.85. According to the manufacturer, a total magnification of 550× could be achieved with the system. Tissue samples of 2.5 × 2.5 cm in size were examined.

### 2.6. Data Collection and Statistical Analysis

The presence of prostate parenchyma and periprostatic tissue (muscles, nerves, adipose tissue, vessels) was documented in an Excel table. Carcinoma formations, atypical/tumor-suspect glands, enlarged nuclei, prominent nucleoli, basal cell loss, and infiltrative growth patterns were recorded. The concluding ratings were provided in a three-grade variable system (0-benign; 1-atypical glands; 2-carcinoma). If present, tumor extent (in percentage) as well as GLEASON score/ISUP prognosis group were documented.

For the histological diagnoses (TH, ChB), tumor manifestations, tumor extent, GLEASON score, ISUP-group and if present perineural, lympho-vascular infiltrates or extra-prostatic manifestations were reported for each biopsy. Inflammatory changes and the presence of pre-neoplastic lesions (Hg-PIN) were also noted.

All statistical analyses were performed manually using Microsoft Excel^TM^. The FCM-ratings (UT, BT) were matched with the diagnoses from conventional histology in error matrices. For the analysis of tumor-diagnoses, FCM-ratings 0 (benign) and 1 (atypical glands) were grouped as not diagnostic for malignancy. Sensitivity, specificity, positive, and negative predictive values were calculated for each evaluator and levels of agreement between FCM and conventional histology were analyzed. The inter-observer variability of the FCM ratings was analyzed in the three-part variable system. Biopsy scans with false-negative tumor-diagnoses and differing ratings were re-examined and morphologically compared to the conventional H&E-slide in order to assess the reasons for the differing opinions.

GLEASON/ISUP-gradings based on FCM diagnoses were analyzed with regard to the ability to detect high-grade cancer. Therefore, the diagnoses were grouped in ISUP grade 1 tumors (GLEASON scores ≤ 6) and ISUP grade > 1 tumors (GLEASON scores ≥ 7). The diagnoses were analyzed in error matrices providing sensitivity, specificity, positive and negative predictive values, and levels of agreement for each pathologist. The biopsies were retrospectively grouped according to the containing ISUP scores in conventional histology and the amount of false ISUP-grades in FCM was evaluated.

The levels of agreement between FCM and conventional histology were measured using the Cohen’s kappa [17]. This coefficient expresses the degree of agreement between two methods in an alternative decision or rating corrected for random matches. The measured agreement (p_0_) of two estimators is related to the coincidentally expected agreement (p_e_). The results can vary between κ = 0 for a purely random match (measured agreement equal to agreement by chance) and κ = 1.0 for a perfect match. The interpretation was based on the Landis and Koch categories (κ < 0.2: poor agreement, κ = 0.2–0.4: fair agreement, κ = 0.4–0.6: moderate agreement, κ = 0.6–0.8: good agreement, κ = 0.8–1.0: very good agreement) [18].

## 3. Results

### 3.1. Biopsies Acquired and Diagnoses in Conventional Histology

A total of 544 biopsies were obtained from 35 patients (Table 1). The selection of the biopsies was very individual and dependent on palpation findings, MRI imaging, PSA value and clinical constellation / family history, resulting in varying numbers of biopsies taken in total (range: 12-35 biopsies). Tumor was present in 19/35 (54.3%) patients. By conventional histology, infiltrates of carcinoma were detectable in 83/544 biopsies (15.3%) altogether, ranging from 1 to 12 biopsies per patient (mean 4.4 ± 3.1).

### 3.2. Manifestations of Tumor in Available FCM Images

Since the sample preparation for FCM took a little longer than the biopsy acquisition, not all biopsies could be scanned for organizational reasons in the operating room, especially in the first patients. In these patients, we focused on target biopsies from MRI lesions. In the later cases, all biopsies were scanned. FCM pictures were acquired in 438 biopsies (mean 12.5 ± 2.8, min 6, max 20). Tumor manifestations were present in 75/438 (17.1%) FCM images of 17/35 patients that could be compared to conventional histology. No FCM images were obtained of incidental tumor manifestations in 8 random biopsies (9.6%) of 4 patients (P01, P03, P14, and P17); these were diagnosed in final conventional histology only. 31/75 (41.3%, 7 patients) FCM-images contained GLEASON 3 + 3 tumor (ISUP 1). 28/75 (37.3%, 6 patients) images showed GLEASON 3 + 4 patterns (ISUP 2). In 2/75 (2.7%, 2 patients) images GLEASON 4 + 3 pattern (ISUP 3) was present. 9/75 (12.0%, 1 patient) images showed GLEASON 4 + 4 tumor (ISUP 4) and 5/75 (6.7%, 1 patient) scans contained GLEASON 5 + 4/5 + 4 patterns (ISUP 5).

### 3.3. Representation of Parenchyma in FCM Images

FCM scans showed a very good agreement of the tissue structures with histological sections allowing direct correlation of these two presentations. Even small groups of glands of less than 1 mm in diameter could be identified unequivocally in both FCM images and FFPE slides (Figure 1). An important difference was seen in the cells of the (hyperplastic) fibromuscular stroma, whose cytoplasm were often shown as bluish rims in fluorescent light (Figure 2a) which in the beginning with only little experience, could be misinterpreted as atypical epithelium.

In FCM images, luminal secretions are well applicable features for the characterization of glandular structures. Atypical luminal secretions of tumor glands are shown as indistinctly contoured crystalloid signals in reflected light (Figure 2c) while corpora amylacea of benign glands appears as homogeneous round structures, which interestingly also give a signal in the fluorescence channel (blue) (Figure 2a). The diagnostically important pigment of the seminal vesicles is shown in the FCM scans intraepithelial deposits of crystalline material in the reflection mode (Figure 2b).

### 3.4. Diagnosis of Malignancy in FCM Images

Histoarchitectural criteria of malignancy [19] could be applied analogously to conventional HE morphology. Infiltrating growth patterns were identifiable by recognition of densely packed micro-glandular structures and caliber-jumps to pre-existing glands (Figure 1). Compared to conventional histology, the basal cells were more difficult to recognize so that their absence was a less reliable criterion for malignancy, especially in small lesions (Figure 3). Cytological criteria of malignancy were very variably represented in the FCM-Images. Nuclear hyperchromasia appeared to be a low reliable criterion for the distinction between tumor cells and glandular epithelia in our FCM images. Nuclear enlargement of tumor cells was well reproduced in the FCM scans, although it differed from case to case. The important criterion of prominent and enlarged nucleoli often was underrepresented in our FCM scans compared with conventional histology, but very helpful when present. Analysis of the biopsies revealed false negative diagnoses of ISUP1 tumors in 10/31 (32.3%) biopsies. Main problem was the distinction from reactive changes, especially in small lesions (sizes of 0.2 to 3 mm). False positive diagnoses were established in 5 biopsies of three patients (P12, P24, P28). Groups of atrophic glands with regenerative nuclear changes or regions with post-atrophic hyperplasia were misinterpreted as well differentiated carcinoma (Figure 4).

Intraoperatively (UT) carcinoma was correctly diagnosed in 64/75 biopsies and 11/75 biopsies were diagnosed as false-negative (sensitivity 85%, specificity 100%, positive predictive value 100%, negative predictive value 97%). In the second evaluation (BT), tumor infiltrates were detected in 62/75 biopsies, false-negative diagnoses were made in 13/75 biopsies and 5 tumor-free biopsies were evaluated as false positive (sensitivity 82.7%, specificity 98.6%, positive predictive value 92.5%, negative predictive value 96.5%). With kappa values of 0.90 and 0.84, the FCM diagnoses showed very good levels of agreement with the gold standard of HE morphology (Table 2).

We found a moderate level of inter-observer agreement (356/438 biopsies, 81.2%, K = 0.56; (Table 3). 285/363 (78.5%) of benign biopsies were consistently rated tumor-free by both pathologists. Non-neoplastic changes in 78/363 (21.5%) of benign biopsies were suspected for malignancy by one or both observers or led to false-positive diagnoses. Tumor was consistently diagnosed by both observers in 59/75 biopsies (78.7%, lesion sizes 0.5–17 mm, 18× ISUP grade1, 26× grade 2, 2× grade 3, 8× grade 4, 5× grade 5). Differing ratings were found in 14/75 (18.7%) biopsies containing tumor (lesion sizes 0.2–12.0 mm, 14× ISUP1, 2× ISUP2). Especially small foci of ISUP grade 1 tumors (lesion sizes 0.2–3 mm) were predominantly grouped false-negative (0) by one rater and suspect for malignancy (1) by the other observer. Consistent false-negative ratings were obtained in 2/75 (2.7%) biopsies (lesion sizes 1.4 mm and 1 mm, 1× ISUP grade 1, 1× ISUP grade 4). The high-grade tumor in these biopsies was not recognizable due to insufficient quality of the scan.

### 3.5. GLEASON Grading in FCM Images

The good reproduction of histological structures allowed GLEASON grading of the tumors in the FCM intraoperatively [14], as this grading system is based on architectural features only. Prognostically unfavorable histological patterns with glandular fusions, glomeruloid and cribriform structures or dedifferentiated carcinoma were present in the FCM scans (Figure 5).

We found moderate levels of agreement (K = 0.56) between FCM and conventional histology for the distinction between ISUP1 tumors and ISUP grades > 1 (Table 4). ISUP grades > 1 were correctly detected in 27/43 biopsies (UT, sensitivity 63%, specificity 100%) respectively 27/41 biopsies (BT, sensitivity 66%, specificity 86%). In summary, 7/7 patients carrying ISUP1 tumor were correctly diagnosed. Tumors requiring therapy (10 patients, 6x ISUP2, 2x ISUP 3, 1x ISUP 4 und 1x ISUP 5) were also reliably diagnosed malignant. 2/6 ISUP2 tumors were falsely grouped as ISUP 1, the other 8 patients (80%) were graded correctly.

Micro-focal ISUP grade 1 lesions were spuriously upgraded as ISUP grade 2 by the less experienced observer in three biopsies of one patient containing ISUP grade > 1 tumor in other biopsies (Table 5). ISUP grades 2 were underrated as ISUP grade 1 in more than half of the biopsies. Of 28 biopsies containing GLEASON3 + 4 (ISUP grade 2) patterns, focally fused glands were correctly detected in 12 (42.8%) resp. 14 (50.0%) FCM scans. In one case, this was due to insufficient image quality. In the remaining cases, fused glandular formations were presented focally and could only be reliably detected in subsequent FFPE processing on deeper serial sections. This led to intraoperative underdiagnosis in 2 (33.3%) of these patients; in the remaining 4/6 patients (66%), higher tumor grades were diagnosed in at least one biopsy. Given technically optimal scans, biopsies containing ISUP grades 3–5 were reliably diagnosed by both pathologists in FCM. In one case of GLEASON 5 pattern, diagnosis of dedifferentiated tumor was established intraoperatively. The final differentiation from a neuroendocrine carcinoma or urothelial carcinoma was established in the paraffin material using additional immunohistology.

### 3.6. Pifalls and Clinically Relevant Non-Neoplastic Changes

FCM images were also well suited for the classification of benign and preneoplastic lesions. In two of the biopsies, small foci of high-grade prostatic intraepithelial neoplasia (HgPIN) were detectable using conventional histology and immunohistology. In the FCM scans (Figure 4c,d) these lesions presented as organoid glands with hyperplastic epithelial lining. Some of the luminal epithelia showed enlarged nuclei with prominent nucleoli. These foci were intraoperatively (UT) rated as normal (0) and as suspect atypical glands (1) in the post hoc examination (BT).

FCM scans also allowed precise statements about inflammatory lesions of the prostate. The extent and the cellular composition of the inflammatory infiltrate can be determined very precisely using the scans. In our scans it was possible to differentiate between acute prostatitis and chronic forms of inflammation (chronic prostatitis and unspecific granulomatous prostatitis, Figure 6).

## 4. Discussion

Our investigations provided comparable results to our colleagues from the working group in Modena [13]. We found similar values of sensitivity (82.7%/85.3%) and specificity (98.6%/100%) for the detection of prostate cancer in prostate biopsies. The most noticeable difference was in the interrater agreement of FCM diagnoses (agreement 95% vs. 82%, kappa-values 0.86 vs. 0.56). This difference reflected the influence of diagnostic experience in the interpretation of smaller lesions (<3 mm in diameter) suspicious for cancer. Small foci of adenocarcinoma, that are often encountered in prostate biopsies, regularly cause confusion with benign processes even in conventional histology [20]. The most useful diagnostic criteria are infiltrative pattern, nuclear enlargement, and prominent nucleoli. Since the previously mentioned criteria showed to be less reliable in the FCM, conventional histology appeared to be superior in these cases. Despite optimal sections and stains, small foci can often only be diagnosed as malignant with help of immunohistology highlighting the lack of the basal layer [19]. It is therefore important that material previously examined with FCM remains suitable for immunohistological examinations.

FCM can be used for intraoperative grading of PCa in prostate biopsies, but it is important to keep in mind its limitations. Larger areas of GLEASON 4 patterns as well as dedifferentiated tumor (GLEASON 5 patterns) were reliably recognizable in the scans. In contrast, a significant proportion of biopsies containing GLEASON 3 + 4 patterns were sometimes underrated as GLEASON 3 + 3 patterns in the FCM. Recognition of GLEASON 4 patterns are challenging even in conventional histology. Diagnostic criteria for these lesions have repeatedly been redefined in ISUP consensus meetings in order to achieve higher levels of reproducibility [21,22,23]. The current definition includes three histological types: cribriform, fused, and poorly formed glands. While the first pattern usually causes little difficulty, detection of small foci of fused or poorly formed glands remains challenging and places high demands on the technical processing of the specimens and on the experience of the pathologists [24]. This is the basis for most of the second opinion requirement, which can lead to a degree of discrepancy close to 45% (K = 0.46) [25]. Although further training effects are to be expected for the FCM, it can be estimated that there remains a proportion of cases in which focal GLEASON 4 patterns within well differentiated carcinoma are not detected in the scans and only be diagnosed in downstream conventional histology.

Despite some minor limitations, FCM appeared to be a useful tool for the intraoperative detection of clinically relevant prostate cancers in prostate biopsies. The morphological heterogeneity of prostate cancer was well represented in the scans allowing reliable grading according to the ISUP system in real time. This material-sparing method conserves the biopsies as unfixed material for further histological, immunohistological, and even molecular analysis. In our preliminary examinations, we found that the pre-treatment for FCM did not alter the feasibility of the material for further FISH examinations and extraction of DNA [26].

Further studies are needed to establish FCM in the routine diagnosis. The intraoperative examination of MRI-targeted biopsies from suspicious lesions may lead to a more effective detection of clinically significant cancers [27]. It should be examined in larger series, to what extent this approach can contribute to the reduction in the total number of biopsies needed and complication rates. The costs saved in this way can be weighed against the acquisition costs of the device.

Over the past decade, the understanding of the phenotypic and molecular heterogeneity in prostate cancer has matured indicating changes in future therapeutic and diagnostic strategies [7]. Much of the complexity of primary prostate cancer diagnosis is rooted in the multifocal nature of the disease. More than 80% of primary prostate cancers show topographically and morphologically distinct tumor foci [28]. Numerous molecular investigations found unique non-overlapping mutation profiles, suggesting that these tumors arise independently and follow separate evolutionary trajectories [29,30]. Thus, any given patient can harbor more than one genomically and phenotypically distinct prostate cancer [31]. Remarkably, whole-exome and whole-genome sequencing studies have demonstrated that anatomically distinct distant metastases share a large number of genomic alterations, confirming that most likely one single clone in the primary tumor gives rise to all distant metastases and that even small well-differentiated lesions can metastasize [32]. Genomic alterations that are associated with initial clinical responses to targeted therapies, such as mutations in DNA repair genes (including BRCA2 and ATM) and mismatch repair genes, have been shown to be truncal and shared between different metastatic sites [33]. It can be estimated that the determination of these markers in obtained tumor tissues will play an important role in the clinical management of metastatic prostate cancer in the future.

To assess the high levels of clonal heterogeneity in future patients, primary tumor samples need to be selected carefully for genomic and epigenetic studies and—if necessary—multi-regional sampling or direct biopsy of metastatic lesions need to be performed. A spectrum of in situ approaches, such as fluorescence in situ hybridization (FISH) and immunohistochemistry have improved the ability to highlight clonally distinct tumor cell populations. FISH-analysis of TMPRSS2-ERG rearrangements and/or immunostaining for PTEN, SPINK1, p53, and RB1 ensure rapid and robust results and have the advantage to be more easily embedded in existing diagnostic workflows [34] than parallel sequencing approaches.

Taking this into account, the possibility for real-time examinations of unfixed diagnostic biopsies and surgical specimens makes the ex vivo FCM an interesting mosaic for future multi-disciplinary diagnostic approaches for prostate cancer. FCM represents a step forward toward digitalized pathology, as the specimen preparation is simple and it provides digital images that enable online remote pathological interpretations. Furthermore, large series of digital FCM images might be a basis for building neural networks in the future that contain data from conventional histology, immunohistology, molecular analyzes, MRI findings, and clinical presentations as well as clinical outcome.

The main advantage of FCM compared to conventional histology is primarily time. The examinations can be performed in the operating room and provide rapid feedback to the surgeons. The first study demonstrated a good feasibility for the intraoperative assessment of surgical margins during radical prostatectomy as an alternative to frozen sections [35]. In pre-therapeutic settings, intraoperative feedback about presence, extent, and grading of tumor in acquired biopsy specimens might contribute to a reduction of biopsies needed in total, when the biopsy results are correlated to MRI findings.

A second advantage of FCM is the preservation of the specimens as unfixed material for downstream immunohistological and molecular examinations. Intraoperative assessment of tumor spread based on mapping biopsies in real-time correlation with MRI-findings, combined with the possibility to perform fast molecular analysis in FISH approaches mentioned above might lead to improved targeting of ablation zones of focal treatments and so could raise their efficacy.

Furthermore, FCM might become an important tool in pathology laboratories for bio-banking of PCa samples. Recent data show that only a small part of the relevant mutations of metastatic tumors are represented in the biopsy specimens [36]. FCM enables ISUP-specific cataloging of the native tissue material, which could contribute to further studies on tumor heterogeneity. With regard to targeted therapies, the search for actionable molecular alterations should be performed in prostatectomy specimens. Using Ex vivo FCM, it should be possible to localize topographically and morphologically distinct tumor foci in native prostatectomy specimens and separately store these tumors as frozen material without loss.

## 5. Conclusions

Ex vivo FCM is a feasible tool for the examination of prostate biopsies enabling the diagnosis and grading of prostate cancer in real-time. Rapid feedback to the examiner might lead to intraoperative adaptions of the biopsy strategy and a reduction of biopsies needed in the individual case. Tumors in need of intervention can already be identified intraoperatively, so that the affected patients can be informed about the further procedure immediately after the biopsy has been taken. The efficiency of future focal therapies could be increased through the rapid feedback.

The specimens are preserved as unfixed material for downstream immunohistological and molecular examinations. Therefore, FCM might become an important tool in pathology laboratories for bio-banking of PCa samples.

## Figures and Tables

**Figure 1 cancers-13-05685-f001:**
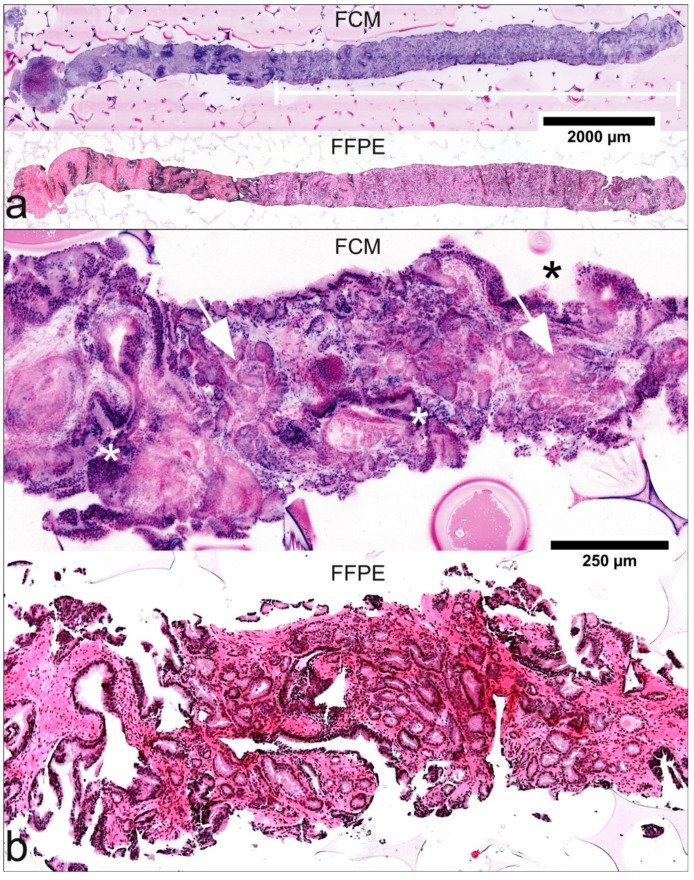
Histological criteria of malignancy in FCM-images and FFPE-processed slides. (**a**) A biopsy of 12-mm length (P28-16) in low magnification. Pre-existing prostatic tissue shows peri-glandular inflammation. The white bar marks well differentiated tumor glands of 7.5 mm length (GLEASON Score 3 + 3; infiltration grade 62.5%). (**b**) FCM-scan and FFPE-processed material of another biopsy (P01-08) show features of infiltrative growth pattern with variable spacing between glands and changing calibers of pre-existing (*) and neoplastic glands (arrows) in higher magnification.

**Figure 2 cancers-13-05685-f002:**
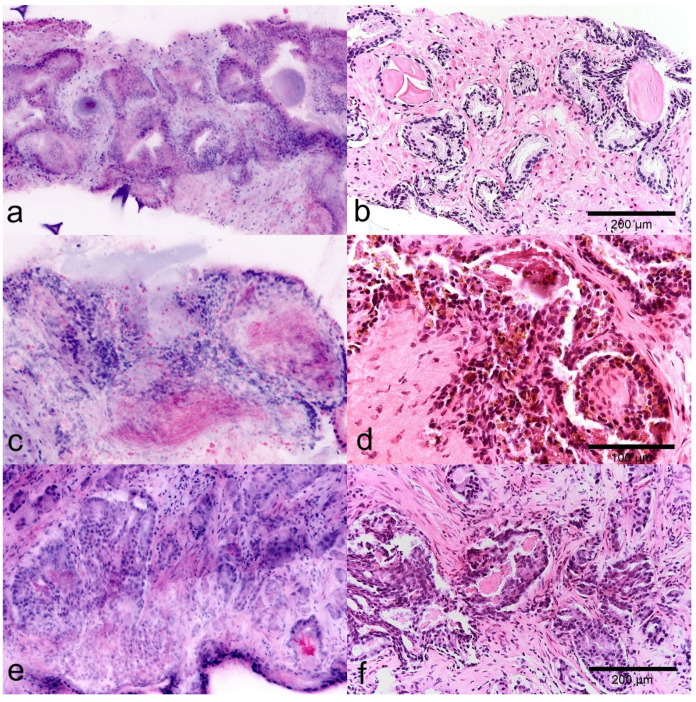
Diagnostic use of glandular secretions in FCM images (left) and FFPE slides (right). (**a**,**b**) Normal prostatic tissue with typical glands. Note the Corpora amylacea in the lumina of the organoid glands. (**c**,**d**) Parenchyma of the seminal vesicle. Diagnostic pigment is represented as intraepithelial granular material in the reflected light (red). (**e**,**f**) Atypical crystalloid secretions in the lumina of tumorous glands are detected as granular material in the lumina in reflected mode (red).

**Figure 3 cancers-13-05685-f003:**
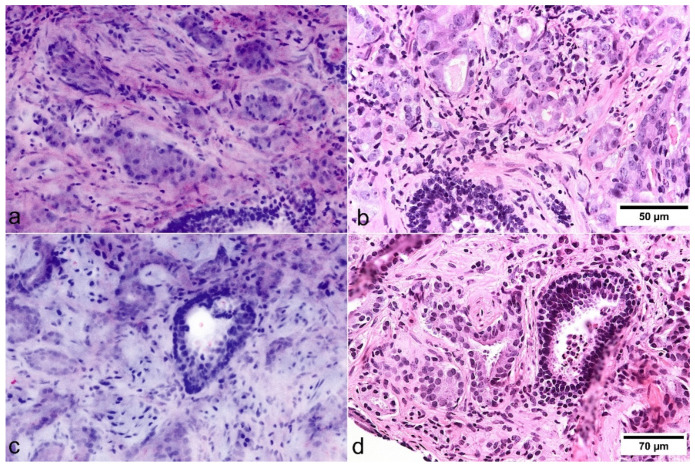
Cytologic criteria of prostate carcinoma in FCM images (left) and FFPE slides (right). (**a**,**b**) The basal cell layer can be identified in pre-existing benign glands. In this case, nuclear enlargement and prominent nucleoli could be easily identified in FCM scans. (**c**,**d**) In this case, nucleolar enlargement and prominent nucleoli are not presented as distinctly in the FCM as in the HE slides.

**Figure 4 cancers-13-05685-f004:**
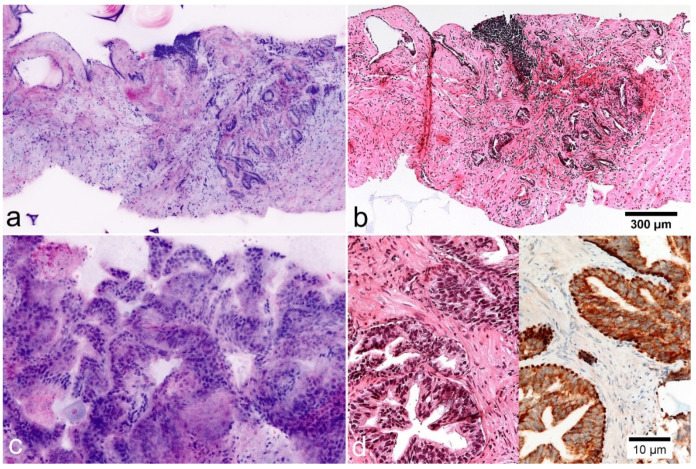
Mimickers of prostate carcinoma in FCM images (left) and FFPE morphology. (**a**,**b**) Postatrophic hyperplasia. Clusters of small glands with hyperplastic stroma and chronic inflammation. The basal cell layer is not visible in the images. (**c**,**d**) High-grade prostatic intraepithelial neoplasia (HgPIN). Organoid glands lined out by atypical epithelium with enlarged nuclei and prominent nucleoli. Immunostaining for P504S shows cytoplasmatic expression in the luminal cells. p40 shows preserved basal cells.

**Figure 5 cancers-13-05685-f005:**
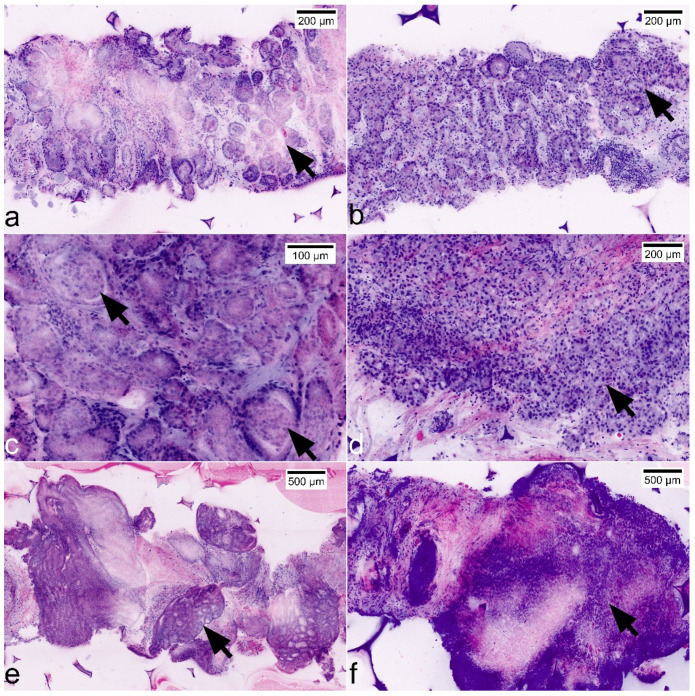
Representation of GLEASON patterns and ISUP grades in FCM scans. (**a**) GLEASON pattern 3 + 3 (ISUP grade 1): non-fused micro-acinar glands; note the atypical secretions in the lumina (arrow). (**b**) GLEASON pattern 3 + 4 (ISUP grade 2): evidence of focal fusions of atypical glands (arrow). (**c**) GLEASON pattern 3 + 4 (ISUP grade 2): note focal glomeruloid formations (arrows) that are included in GLEASON 4-pattern since the consensus classification in 2016. (**d**) GLEASON pattern 4 + 3 (ISUP grade 3): Dominance of fused glands (arrow), some micro-acinar formations in the periphery. (**e**) GLEASON-pattern 4 + 4 (ISUP grade 4): large complex cribriform tumor formations. (**f**) GLEASON-pattern 5 + 5 (ISUP grade 5): solid formations of highly atypical tumor cells.

**Figure 6 cancers-13-05685-f006:**
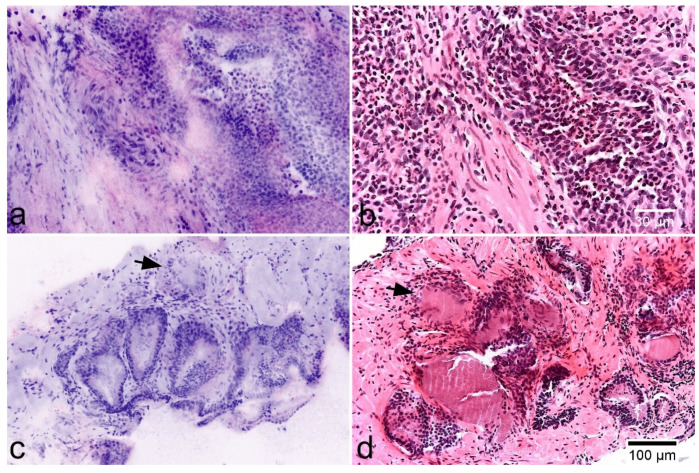
Inflammatory lesions of the prostate. (**a**,**b**) Acute prostatitis in FCM scans (left) and HE-morphology. Note a dense granulocytic infiltrate in the stroma and epithelium. (**c**,**d**) Non-specific granulomatous prostatitis showing enlarged glands and a periglandular inflammatory infiltrate of lymphocytes and giant cells (arrows).

**Table 1 cancers-13-05685-t001:** Clinical data, number of biopsies taken and examined in FCM, tumor manifestations.

Patient	Age	PSA	PIRADS	Indication	Biopsies/Tumor			FCM Images/Tumor			Gleason	ISUP
P01	65	6.5	4	AS	24	/	7	12	/	6	3 + 3	1
P02	68	7.8	4	Pre	35	/	0	17	/	0	-	-
P03	77	3.2	3	AS	24	/	7	10	/	2	4 + 3	3
P04	66	16	5	Pre	30	/	0	10	/	0	-	-
P05	79	11.6	5	Pre	12	/	1	12	/	1	3 + 3	1
P06	60	8	n.o.	AS	12	/	0	12	/	0	-	-
P07	57	9	5	Pre	12	/	5	12	/	5	3 + 4	2
P08	49	1.1	5	Pre	12	/	0	12	/	0	-	-
P09	57	6.4	2	Pre	12	/	5	12	/	5	4 + 3	3
P10	77	7.8	n.o.	Pre	12	/	0	12	/	0	-	-
P11	59	9.6	4	Pre	12	/	3	12	/	3	3 + 4	2
P12	79	55	5	Pre	14	/	7	14	/	7	5 + 4	5
P13	64	11.8	3	Pre	12	/	0	6	/	0	-	-
P14	64	6.6	4	Pre	12	/	1	6	/	0	3 + 3	1
P15	61	3.55	3	Pre	12	/	0	9	/	0	-	-
P16	66	18.6	4	Pre	20	/	0	14	/	0	-	-
P17	78	4.68	4	Pre	21	/	1	11	/	0	3 + 3	1
P18	72	16.4	3	Pre	14	/	0	12	/	0	-	-
P19	74	14.55	3	Pre	14	/	0	7	/	0	-	-
P20	58	5.96	5	AS	12	/	1	12	/	1	3 + 3	1
P21	66	5.4	3	Pre	15	/	0	14	/	0	-	-
P22	66	9.92	5	Pre	13	/	9	13	/	9	3 + 4	2
P23	73	8.7	5	Pre	12	/	5	12	/	5	4 + 4	4
P24	69	12.9	4	Pre	12	/	5	12	/	5	3 + 3	1
P25	77	18	5	Pre	14	/	3	14	/	3	3 + 4	2
P26	61	33	5	Pre	14	/	12	14	/	12	3 + 4	2
P27	64	32.7	5	Pre	20	/	3	20	/	3	3 + 4	2
P28	52	6.39	4	Pre	16	/	6	16	/	6	3 + 3	1
P29	63	5.89	5	Pre	14	/	1	14	/	1	3 + 3	1
P30	61	3.31	5	Pre	15	/	0	15	/	0	-	-
P31	64	7.28	5	AS	12	/	1	12	/	1	3 + 3	1
P32	63	4.41	4	Pre	13	/	0	13	/	0	-	-
P33	69	6.86	4	Pre	15	/	0	15	/	0	-	-
P34	68	7.83	4	AS	16	/	0	16	/	0	-	-
P35	55	3.81	5	Pre	15	/	0	14	/	0	-	-
**total**					**544**	**/**	**83**	**438**	**/**	**75**		

Age, Serum PSA and PIRADS is shown for each patient. AS: follow-up-biopsies under active surveillance; Pre: pre-therapeutic biopsies. The number of biopsies obtained in total as well as the numbers of acquired FCM images are listed. For patients with evidence of tumor, GLEASON Score, ISUP-Grade and the number of biopsies/FCM images containing tumor are shown. In two patients (P14, P17) carcinoma was not represented in FCM scans.

**Table 2 cancers-13-05685-t002:** Level of agreement between diagnoses in FCM images (UT, BT) and FFPE morphology (TH/ChB).

			FCM-Ratings (UT)			FCM-Ratings (BT)		
			0	1	2	0	1	2
FFPE TH/ChB	Benign (n = 363)		*343*	*20*	**0**	*295*	*63*	**5**
			**363**			**358**		
	Carcinoma (n = 75)		*6*	*5*	**64**	*4*	*9*	**62**
			**11**			**13**		
	Total	438	374		64	371		67
Sensitivity			85.3%			89.3%		
Specificity			100%			98.6%		
Positive predictive value			100%			93.1%		
Negative predictive value			97.1%			96.5%		
Cohen’s Kappa			0.90			0.85		
Level of agreement *			very good			very good		

FCM-ratings (0 = no tumor; 1 = suspicious for tumor; 2 = presence of carcinoma) compared to the final histological diagnoses in an error matrix.

**Table 3 cancers-13-05685-t003:** Interobserver variability of FCM diagnoses between intraoperative (UT) and post-hoc (BT) investigation.

		FCM (UT)			
		0	1	2	Σ
**FCM (BT)**	**0**	287	11	1	299
	**1**	58	10	4	72
	**2**	4	4	59	67
	**Σ**	349	25	64	438
**Agreement**		287	10	59	356
**Kappa**		0.56			
**Level of agreement**		moderate			

FCM ratings (0 = no tumor; 1 = suspicious for tumor; 2 = presence of carcinoma) were compared between the two raters (UT, BT). High levels of interrater agreement were found for tumor-diagnoses (2) and for diagnoses of tumor-free biopsies. There were clear differences in the classification of suspicious glands (1). UT classified a lower rate of biopsies as suspicious (5.7%) than BT (16.4%) representing different levels of diagnostic experience.

**Table 4 cancers-13-05685-t004:** Comparison of ISUP grades in FCM scans and conventional histology.

		FCM (UT)		FCM (BT)	
		ISUP Grade 1	ISUP Grade > 1	ISUP Grade 1	ISUP Grade > 1
**FFPE (TH/ChB)**	**IUSP Grade 1**	21	0	18	3
	**ISUP Grade > 1**	16	27	14	27
**Biopsies (total)**		64		62	
**Sensitivity**		63%		66%	
**Specificity**		100%		86%	
**Positive predictive value**		100%		90%	
**Negative predictive value**		57%		56%	
**Cohen’s Kappa**		0.52		0.46	
**Level of agreement**		moderate		moderate	

Diagnoses of GLEASON patterns in FCM (UT, BT) and conventional histology (FFPE TH/ChB) were grouped in ISUP grade 1 (GLEASON ≤ 6) and ISUP grades > 1 (GLEASON ≥ 7) and analyzed an error matrix. There is a moderate level of agreement of FCM with the conventional histology.

**Table 5 cancers-13-05685-t005:** Limitations of tumor diagnoses and ISUP grading in FCM scans.

FFPE (TH/ChB)		FCM (UT)				FCM (BT)			
ISUP	n	False Neg		False ISUP		False Neg		False ISUP	
1	31	10	32%	0	0%	10	32%	3	10%
2	28	0	0%	16	57%	2	7%	14	50%
3	2	0	0%	0	0%	0	0%	0	0%
4	9	1	11%	0	0%	1	11%	0	0%
5	5	0	0%	0	0%	0	0%	0	0.0%

Individual analysis in regard to detected ISUP grades in the biopsies revealed high levels of false negative findings for ISUP grade 1 tumors. ISUP grade 2 tumors were often under-graded as grade 1 lesions. ISUP grades 3–5 were reliably diagnosed and correctly grouped as high-grade tumor.

## Data Availability

All data generated or analyzed during this study are included in this published article.

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
