# Peer review of "Diagnostic Performance of Ex Vivo Fluorescence Confocal Microscopy in the Assessment of Diagnostic Biopsies of the Prostate"

_cancers, 2021, doi:10.3390/cancers13225685_

Round 1
Reviewer 1 Report
The work is original and interesting.
The number of patients is not relevant but considering the single samples it becomes adequate.
It needs another statistical valutation with a suitable software before the publication.
Author Response
We chose the statistical methods on the basis of results from previous working groups in order to provide comparable data as far as possible. These are established parameters for evaluating test procedures. In our opinion, these statistical values can be calculated manually without much effort.
Reviewer 2 Report
The authors tested the FCM for prostate cancer diagnosis. The topic is interesting and paper is well drafted.
I just have few minor comments:
- can authors share the vision on: at what performance level, will the FCM be able to introduce to clinical practice? Will it be more suitable for certain subgroup of patients?
- An exciting future direction is to test if deep learning can help improve the disease diagnosis based on FCM.
Author Response
I just have few minor comments:
- can authors share the vision on: at what performance level, will the FCM be able to introduce to clinical practice? Will it be more suitable for certain subgroup of patients?
*****
Answer:
The retrospective analysis of our data suggested, in line with the recent literature, that tumor diseases requiring intervention could already be detected by analyzing the target biopsies from lesions suspect for malignancy in the MRI (PIRADS4 and -5) using FCM.
We actually examine in a new series of patients naïve of Prostate Cancer to what extent the intraoperative FCM analysis of these target biopsies is able to identify patients in need of intervention. Prompt diagnosis would make patient management easier. The feedback in the operating room might result in a reduction in the number of biopsies required und thus lead to a reduced rate of complications.
*****
- An exciting future direction is to test if deep learning can help improve the disease diagnosis based on FCM.
*****
Answer:
We have already contacted a technical university with this question. FCM represents a step forward towards digitalized pathology, as the specimen preparation is simple and it provides digital images that enable online remote pathological interpretations. Furthermore, large series of digital FCM images might be a basis for building a neural network in the future that contains data from conventional histology, immunohistology, molecular analyzes, MRI findings and clinical presentations as well as clinical outcome.
*****

Reviewer 3 Report
Titze et al. evaluate the diagnostic potential fluorescence confocal microscopy (FCM) for the detection and assessment of prostate cancer in biopsies intraoperatively. Prostate biopsies of 35 patients were stained and imaged by FCM before being fixed and prepared for conventional histology. The assessment based on FCM was validated by the diagnosis based on histology, showing moderate agreement for minor cancers and good agreement for more serious cancers.
Whereas the study was designed in principle for a direct comparison of FCM versus histology of the same tissue, the direct comparison seems hampered by differences in the analyzed tissue sections, because the presented tissue structures rarely match for the two imaging modalities. If the fluorescence staining survives the histology protocol, reimaging the histology sections by FCM a posteriori may minimize structural differences and improve the validation.
Please indicate the magnified region of figure 1a shown in figure 1b and 1c.
The authors chose Cohen’s kappa with the interpretation by Landis and Koch (lines 159-162) for their assessment of the diagnostic match between imaging modalities and pathologists. Cohen’s kappa yields a summary value <= 1 whose interpretation is debatable except for near perfect match (kappa > 0.9). Positive and negative predictive values may be criteria that are more suitable; they are discussed in the text.
Table 1 lists the clinical data for all biopsies taken. For patients P01–P04 and P13–P19, FCM was analyzed for a third to half of the biopsies. For patients P21 and P35, FCM missed one biopsy. FCM and histology showed a perfect match for the other patients. Please explain why FCM was taken for some biopsies only in some patients.
As patients P01 and P03 are listed by 6 and 2 tumor detections in FCM images, I do not understand the meaning of “No FCM images were obtained of incidental tumor manifestations in 8 random biopsies (9,6 %) of 4 patients (P01, P03, P14 and P17); these were diagnosed in final conventional histology only.” on lines 177–179.
The FCM and histology images may be valuable for other pathologists’ education and training. Would the authors make the images and the assessment of each image available? If so, please upgrade the data availability statement.
Before publication, please consider the following minor edits. Thank you.
- 488nm wavelength is blue light and 785nm near-infrared light (line 120).
- Please correct the format of the second line in table 1.
- A scale bar may be provided for all images.
- Please correct the sensitivity and specificity values for BT on line 288.
- Please round percentages in tables 4 and 5 to integers because the denominators are < 100.
- Please check “Despite some minor these limitations,” on line 365.
Author Response
Titze et al. evaluate the diagnostic potential fluorescence confocal microscopy (FCM) for the detection and assessment of prostate cancer in biopsies intraoperatively. Prostate biopsies of 35 patients were stained and imaged by FCM before being fixed and prepared for conventional histology. The assessment based on FCM was validated by the diagnosis based on histology, showing moderate agreement for minor cancers and good agreement for more serious cancers.
Whereas the study was designed in principle for a direct comparison of FCM versus histology of the same tissue, the direct comparison seems hampered by differences in the analyzed tissue sections, because the presented tissue structures rarely match for the two imaging modalities. If the fluorescence staining survives the histology protocol, reimaging the histology sections by FCM a posteriori may minimize structural differences and improve the validation.
Please indicate the magnified region of figure 1a shown in figure 1b and 1c.
*****
These are images from different biopsies. The statement should be the histological diagnostic criteria. I have revised the legend of the figure accordingly.
*****
The authors chose Cohen’s kappa with the interpretation by Landis and Koch (lines 159-162) for their assessment of the diagnostic match between imaging modalities and pathologists. Cohen’s kappa yields a summary value <= 1 whose interpretation is debatable except for near perfect match (kappa > 0.9). Positive and negative predictive values may be criteria that are more suitable; they are discussed in the text.
Table 1 lists the clinical data for all biopsies taken. For patients P01–P04 and P13–P19, FCM was analyzed for a third to half of the biopsies. For patients P21 and P35, FCM missed one biopsy. FCM and histology showed a perfect match for the other patients. Please explain why FCM was taken for some biopsies only in some patients.
*****
Unfortunately, the biopsies were taken by several surgeons and there was no standardized biopsy protocoll. The selection of the biopsies was very individual and dependent on palpation findings, MRI imaging, PSA value and clinical constellation / family history.
The decision to go for an FCM scan was very much dependent on external factors in the operating room. We tried to a systematic selection (e.g. every 2nd biopsy, see cases P13 + P14). Finally, we focused on the target biopsies from MRI lesions or scanned all biopsies in the later cases.
******
As patients P01 and P03 are listed by 6 and 2 tumor detections in FCM images, I do not understand the meaning of “No FCM images were obtained of incidental tumor manifestations in 8 random biopsies (9,6 %) of 4 patients (P01, P03, P14 and P17); these were diagnosed in final conventional histology only.” on lines 177–179.
*******
Patient01: 7/24 biopsies in total carried tumor. Tumor was found in 6/12 images. So 12 biopsies were not scanned of which one biopsy contained tumor.
Patient03: 7/24 biopsies in total carried tumor. Tumor was visible in 2/10 images. Another 12 biopsies were not scanned of which 5 biopsies contained tumor.
Patient14: 1/12 biopsies contained tumor. Only every second biopsy was 6 scanned, so we missed one biopsie containing tumor.
Patient17: 1/21 biopsies in total carried tumor. Tumor was found in 0/11 images. So 12 bipsies were not scanned of which one biopsie contained tumor.
So in total, 1 + 5 + 1+ 1 = 8 biopsies containing tumor were not scanned in FCM.
******
The FCM and histology images may be valuable for other pathologists’ education and training. Would the authors make the images and the assessment of each image available? If so, please upgrade the data availability statement.
*****
The image data set can currently not be made publicly available.
******
Before publication, please consider the following minor edits. Thank you.
- 488nm wavelength is blue light and 785nm near-infrared light (line 120). Fixed
- Please correct the format of the second line in table 1.
- A scale bar may be provided for all images. Fixed.
- Please correct the sensitivity and specificity values for BT on line 288. Fixed
- Please round percentages in tables 4 and 5 to integers because the denominators are < 100. Fixed
- Please check “Despite some minor these limitations,” on line 365. Fixed

Reviewer 4 Report
The authors evaluated ex vivo FCM in pathological diagnosis of pre-therapeutic prostate core biopsy specimens. The authors demonstrated its diagnostic performance by comparing standard histopathological diagnosis. This is an interesting study. The technology introduced in the present study is expected to apply to various clinical situations described by the authors. The authors may want to address the following points before publication. 1_ The authors may want to clarify whether their study was prospective or retrospective. 2_ The authors may want to provide information on and discuss cost generated by the novel procedure.Author Response
Ad 1:
It was a prospective study.
Ad 2:
The acquisition costs for the microscope are € 250,000. The additional time of the pathologist in the operating room and the costs of consumables were not systematically recorded in this study. This will be done in the following work, which examines the extent to which intraoperative FCM microscopy of MRI-targeted biopsies from suspicious foci can reduce the number of biopsies and the length of the hospital stay. Costs and benefits can then be compared.